# QRad: Enhancing Radiology Report Generation by Captioning-to-VQA Reframing

## Abstract

Radiology Report Generation using AI has demonstrated significant potential in modern clinical workflows. However, existing approaches have limited clinical utility due to a lack of interactive capabilities and compromised factual reliability because linguistic variations are prevalent in the training data and lead to overfitting. We introduce QRad, a novel approach which reframes radiology report generation from image captioning to a self-directed visual question-answering (Auto-VQA) process. Specifically, we convert radiology reports into question-answer pairs and train our model to first generate the chain of questions and then respond with answers. The answers are concatenated to form the radiology report. Our approach offers three advantages: First, quality is considerably improved because sentence-level linguistic variations (such as the omission or ordering of medical topics) are removed from the answer generation's criterion, allowing the model to focus on factual accuracy rather than presentation style. Second, the model provides an intrinsic VQA capability that enables physicians to interact with the model for details that may have been omitted in the initial output. Third, QRad derives confidence scores from token probabilities through its ability to answer template questions about specific medical conditions, a capability unavailable in previous models, enabling Receiver Operating Characteristic (ROC) based evaluation to facilitate regulatory approvals. Experiments show that QRad outperforms state-of-the-art models with only 13% of their sizes, offering a promising path for clinical adoption and regulatory validation in real-world settings.

## 1 Introduction

Medical imaging plays a crucial role in healthcare diagnostics. However, the worldwide shortage of radiologists poses significant risks to patient care (Ganeshan et al., 2020; Parikh et al., 2020; Cao et al., 2023). Automated radiology report generation using AI has emerged as a promising solution to this challenge, with the potential to reduce radiologist burden to only the most complex cases.

Despite recent advances in radiology report generation, significant gaps remain towards clinical adoption. First, current approaches, which typically follow an image captioning pipeline, struggle with the inherent linguistic uncertainties (Tanno et al., 2025) in radiology reports. Unlike conventional image captioning, radiology reports are longer documents that require precise factual accuracy while exhibiting considerable sentence-level linguistic variation, such as whether a finding is mentioned or omitted, and the order in which medical findings are presented. As a simplified example, if the ground truth has three sentences [A, B, C], a prediction that reorders the same findings (e.g., [C, A, B]) is clinically correct but is unfairly penalized (Huang et al., 2019) by the language modeling loss because it requires exact token-by-token matches [1]. Consequently, models tend to overfit such linguistic variations at the expense of factual accuracy. In report generation datasets such as MIMIC-CXR (Johnson et al., 2023), each training sample contains one or multiple images and an associated text report. Image captioning datasets like COCO (Lin et al., 2014) provide multiple reference texts to capture the linguistic variances, however, this solution is not feasible in the collection of radiology report datasets. Furthermore, conventional approaches that follow a direct image-to-text pipeline (Chaves et al., 2024; Tu et al., 2024; Yang et al., 2024; Chen et al., 2024)

---

[1] Teacher forcing during training may reduce the effect.

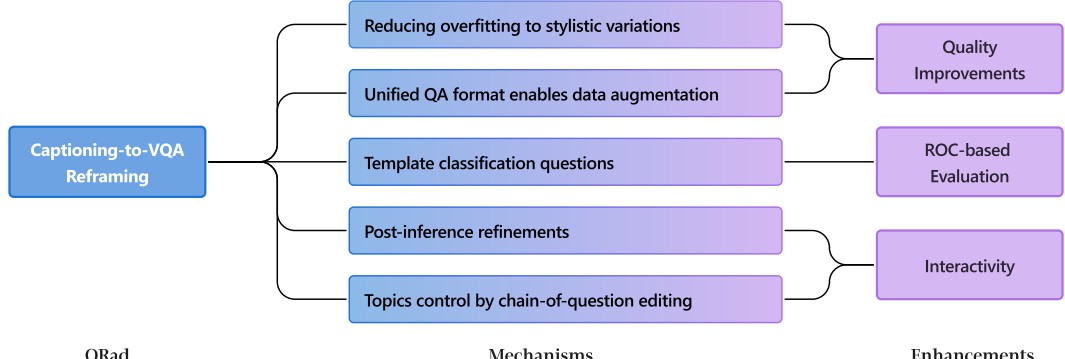

Figure 1: Overview of QRad's captioning-to-VQA reframing approach. This question-driven framework (the "Q" in QRad) enables five mechanisms that collectively enhance clinical utility across three dimensions: report quality, regulatory-required evaluations, and interactivity.

offer no interactive mechanisms, preventing physicians from requesting additional information about specific concerns omitted in the initial output (Pal et al., 2025; Hu et al., 2024).

Second, existing generative models lack the ability to produce continuous, numerical confidence scores for individual medical findings. For clinical utilization of software, FDA device authorization requires generating Receiver Operating Characteristic (ROC) curves and evaluating sensitivity and specificity across clinical applications with differing tolerances for false positives and false negatives (Food, 2007). For example, cancer screening prioritizes high sensitivity to avoid missed cases, while cohort discovery systems for clinical research require high specificity to accurately identify patients meeting strict inclusion criteria and reduce downstream noise. A model that produces confidence scores for requested disease classes can therefore facilitate regulatory approval, moving one step closer to real-world adoption.

To address these challenges, we introduce $Q$Rad, a novel approach that reframes radiology report generation as a self-directed visual question-answering (Auto-VQA) process. $Q$Rad operates in two steps: (1) Question Generation, which produces a chain of relevant clinical questions conditioned on the input radiograph, effectively planning the report's structure; (2) Answer Generation, which answers those questions by examining visual features. The answers are concatenated to form the final report. To facilitate training, we convert reference reports into QA pairs by segmenting each report into contiguous topical spans (answers), and GPT-4o[2] generates a single question that captures each span's topic. This design offers two immediate benefits: it operationalizes Chain-of-Thought (Wei et al., 2022) via explicit planning-and-answering decomposition, and it provides an interactive capability, allowing physicians to request specific information beyond the initial report by editing or issuing follow-up questions—a feature unavailable in previous single-step models.

Formally, traditional approaches model report generation as $Y = f(I)$, where an image $I$ directly maps to a report $Y$. Due to valid linguistic variations in the ground truth such as reordering of sentences, this formulation suffers from a one-to-many mapping from $I$ to multiple valid $Y$, causing the learning process to overfit to surface phrasing at the expense of clinical accuracy. This limitation arises from the language modeling loss which treats each token equally, allowing the model to shortcut by producing a radiology report that achieves linguistic overlap with the ground truth on non-factual tokens while differing in key tokens that determine factual accuracy, such as presence/absence, severity, and location. $Q$Rad reframes the process as $Y = f_A(I, Q); , Q = f_Q(I)$. By providing the Answer Generator $f_A$ with a question $Q$, we explicitly demand the model to state a diagnosis for the clinical topic. The ground truth for $f_A$ is a single-sentence topical span (answer), reducing the space of linguistic variations and focusing on factual accuracy. The Question Generator $f_Q$ captures linguistic variability—even when it produces questions that differ from the training data, these tend to be clinically valid variations that preserve diagnostic utility. In essence, we isolate linguistic variability in $f_Q$ and concentrate factual supervision in $f_A$. Moreover, the VQA reframing allows us to augment the training data with additional image classification questions; in these cases, the

---

[2]We use a private, in-house deployment to satisfy data-usage requirements. The labeled data will be released.

ground truth is a single `Yes`/`No` token which further reduces linguistic variability and concentrates supervision on diagnostic accuracy.

Furthermore, typical regulatory processes (e.g., FDA approval) require ROC-based validation, which depends on class probabilities like those produced by perception models. $Q$Rad bridges this gap via *closed-vocabulary* VQA: for each predefined disease class, we pose a binary, template-based query (e.g., "`Is this image classified as [CLASS]? (yes/no)`"), extract the token logits for pre-defined answers {`Yes, No`}, and compute the softmax as class probabilities. In contrast, conventional report-generation models emit free-form sentences that may mention multiple diseases or omit a disease entirely, so token-level probabilities are not class-specific and cannot serve as per-class confidences to support ROC analysis. Meanwhile, image classifiers do not produce open-vocabulary reports that describe medical findings with flexibility. Our VQA reframing approach unifies both regimes, providing open-vocabulary narratives and closed-vocabulary class probabilities within a single backbone to support ROC/AUC analysis, offering a practical path toward regulatory clearance and real-world adoption.

In summary, we propose $Q$Rad, a captioning-to-VQA reframing approach that addresses key limitations in radiology report generation. As illustrated in Figure 1, our question-driven framework enables five core mechanisms that collectively enhance clinical utility across three critical dimensions: improving report quality by reducing overfitting to stylistic variations, enabling ROC-based evaluation through quantitative confidence scores, and providing interactivity enhancements via post-inference refinements and topic control. Experiments show that $Q$Rad outperforms state-of-the-art models (Zhang et al., 2025a; Zhou et al., 2024; Chen et al., 2024; Chaves et al., 2024) while using only 13% of their model size.

## 2 RELATED WORK

### 2.1 IMAGE CAPTIONING

Image captioning aims to generate a sentence that describes a given image. The latest work benefits from large scale vision-language pre-training (Chen et al., 2020a; Dou et al., 2021; Wang et al., 2021; Kim et al., 2021). Encoder-decoder architectures (Li et al., 2023; Wang et al., 2022; Nguyen et al., 2022) provide a unified implementation for various vision-language tasks.

While many radiology report generation methods are based on image captioning (Cornia et al., 2020; Vinyals et al., 2015; Xu et al., 2015; You et al., 2016), there are key differences in the tasks including (1) radiology reports are much longer than generic image captions such as those in COCO Captions (Lin et al., 2014), and have multiple sentences covering different medical topics; (2) factual correctness is critical for radiology reports, which requires close examination of fine visual details; (3) image captioning datasets may provide multiple ground truths per image to capture linguistic variations, however, this is not available in radiology report datasets.

### 2.2 RADIOLOGY REPORT GENERATION

Chest X-ray radiology reports lack a standardized order for presenting medical findings (Burbridge, 2017). For instance, the inside-out order (Smithuis & Otto, 2022) and the ABCDE order (each letter represents an anatomical region) (Lopez-Cardona, 2023) are two approaches from clinical guidelines. Additionally, medical conditions can be omitted from the report (Irvin et al., 2019). These valid linguistic variations lead to Loss-Metric mismatch problems, creating challenges for both training and evaluation (Gu et al., 2018b; Yi et al., 2020; Gu et al., 2018a). Existing state-of-the-art methods use the original radiology reports as supervision and train the models in an image captioning setup, differing primarily in datasets, architectures, and pretraining/fine-tuning regimes. Early work connects a frozen image encoder to a pre-trained language model such as LLaMA (Li et al., 2024; Chaves et al., 2024) and later work explores mimicking clinical setups (Bannur et al., 2024) and leverages pre-training and fine-tuning techniques (Yang et al., 2024; Nath et al., 2024; Burbridge, 2017). Previous studies also demonstrated the value of generating reports using a two-step approach (Nooralahzadeh et al., 2021; Liu et al., 2019; Yan et al., 2023), which are conceptually similar to ours. However, due to the absence of sentence-level concept labels, these methods rely on unsupervised topics or proxy targets.

Specifically, Liu et al. (2019) adopts a hierarchical framework that predicts sentence-level topics as the first step. However, their topic generation module is not supervised with any labels, leaving uncertainty in their actual meaning. Nooralahzadeh et al. (2021) first generates high-level context sentences and then refines them into the reports. The first step is trained to generate medical keywords per sentence extracted with a text processing model. We differ from them in the supervision of the first step. Yan et al. (2023) replaces full reports with serialized RadGraph representations (entities and attributes) as supervision, thereby filtering out non-semantic words. In contrast, *Q*Rad addresses sentence-level style variations, such as omission and reordering of findings, which RadGraph-based supervision still encodes.

Like most existing research, *Q*Rad aims to generate free-text reports, different from the structured report generation task (Delbrouck et al., 2025; Pellegrini et al., 2023a) which standardizes the format of radiology reports to reduce the linguistic variance. Besides, our Auto-VQA process is different from the conventional VQA setup in existing work (Özdemir & Akagündüz, 2024; Zhang et al., 2025b; Hu et al., 2022; Serra et al., 2025) in that both the questions and answers are predicted by our model, where the questions are for planning the structure of the report for each image input.

## 3 METHOD: REFRAMING LONG TEXT GENERATION TO AUTO-VQA

Conventional approaches to long text generation from visual inputs frame the task as direct image-to-text mapping i.e., image captioning. As valid linguistic variations are prevalent in radiology reports, amplified by their length, factual accuracy is hindered when the model attempts to overfit the linguistic variations. We propose a general approach that reframes long text generation into Auto-VQA, a self-directed visual question-answering process where the self-generated questions serve as an explicit plan akin to chain-of-thought (Wei et al., 2022) models.

The proposed Captioning-to-VQA reframing method is generalizable to different model architectures. In our experiments, it effectively elevates the performance of a small model to match those 10X larger. In this section, we demonstrate our method with MIMIC-CXR (Johnson et al., 2023), one of the largest radiology report datasets that are publically available.

### 3.1 DATASET PREPARATION

*Q*Rad requires two types of question-answering (QA) datasets, including a report generation QA dataset converted from the image-report dataset, and an image classification QA dataset converted from image-class labels. Compared to using the original full reports as supervision, the first dataset reduces linguistic variations at the sentence-level (such as the omission and ordering of sentences), while the second data, being closed-vocabulary (the answers being {Yes, No}), further reduces linguistic variations at the phrase level.

#### 3.1.1 REPORT GENERATION QUESTION-ANSWER PAIRS (OPEN VOCABULARY)

To generate these datasets from image captions, we use an LLM[3] to split the reports into sentence groups. Consecutive sentences in a report covering the same topic are treated as a cohesive unit. Then, we use each sentence group as an answer, and compose a corresponding question with the LLM. As shown in Figure 2, when generating the questions, we instruct the questions to be precise enough to indicate the topics while not being too specific to leak the answer.

For the MIMIC-CXR (Johnson et al., 2019) dataset, we generated a total of 818,867 question-answer pairs across all radiology studies. There are 110,959 unique questions (based on string matches, not semantic similarity). 91.3% of the reports have no more than 5 sentences, and 99.4% of the reports have no more than 8 sentences. Typical answers contain only one sentence.

#### 3.1.2 IMAGE CLASSIFICATION QUESTION-ANSWER PAIRS (CLOSED VOCABULARY)

One benefit of our Captioning-to-VQA reframing is the ability to unify different supervisions into the same VQA format, allowing our model to seamlessly learn from both kinds of annotations

---

[3]We use a private, in-house deployment of GPT-4o (Hurst et al., 2024) to ensure compliance with the dataset usage requirements. The data processing does not assume a particular LLM.

```
• [Q1] "What type of view is used in the chest X-ray?"
  [A1] "Single AP view of the chest provided."

• [Q2] "Are there any support devices visible?"
  [A2] "An endotracheal tube ends 2.0 cm above the Carina.  A
        transesophageal tube courses below the level of the
        diaphragm, however the tip cannot be visualized."

• [Q3] "What is the condition of the lung volumes and clarity?"
  [A3] "Lung volumes are low, however grossly clear."

• [Q4] "Is there any atelectasis?"
  [A4] "Bibasilar atelectasis is moderately increased."

• [Q5] "Are there signs of pleural effusion or pneumothorax?"
  [A5] "No pleural effusion or pneumothorax."
```

Figure 2: Example of the converted report generation QA dataset. We show the first five sentences from a radiology report, where $Q_i$ and $A_i$ are the $i^{th}$ question and answer, respectively.

to achieve superior performance. Here we augment image-report data with image-class labels. Specifically, in addition to the report generation QA pairs, we convert image class labels (obtained from VisualCheXbert (Jain et al., 2021b)) into the VQA format. This integration not only enhances our model's image understanding capabilities but also improves its ability to handle diverse input questions while providing a natural mechanism for confidence score extraction.

```
• [Q1] "Is this image classified as cardiomegaly? (yes/no)"
  [A1] "Yes"

• [Q2] "Does this chest X-ray demonstrate edema? (yes/no)"
  [A2] "No"

• [Q3] "Is pleural effusion evident in this chest X-ray? (yes/no)"
  [A3] "Yes"

• [Q4] "Does this radiograph indicate pneumothorax? (yes/no)"
  [A4] "No"

• [Q5] "Does this chest X-ray reveal support devices? (yes/no)"
  [A5] "No"
```

Figure 3: Example of question-answer pairs converted from image classification labels. The questions are formulated using question templates and pre-defined class names, with a "(yes/no)" suffix that distinguishes them from report generation QA pairs and indicates a single-token binary answer is expected.

As shown in Figure 3, classification labels are transformed to closed-vocabulary QA pairs using the 14 categories from CheXpert (Irvin et al., 2019). Questions are constructed by randomly sampling from a template pool. The closed-vocabulary nature of these QA pairs focuses on training the model's image classification capabilities like an image classifier. When training on such datasets, the model gets no reward for writing a full sentence that has token-wise overlap with the ground truth sentence but is factually incorrect.

## 3.2 Auto-VQA Pipeline and Model Architecture

QRad decomposes the traditional image-to-text generation task from $Y = f(I)$ into two distinct components: a Question Generation Module $Q = f_Q(I)$ and an Answer Generation Module $Y = f_A(I, Q)$, where $I, Q, Y$ denote the input image, questions, and answers (sentences in the report),

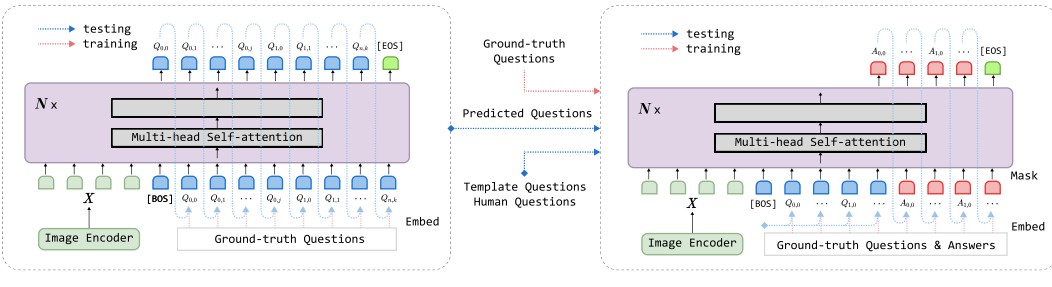

Figure 4: *Q*Rad's Auto-VQA pipeline. The Question Generator predicts a sequence of questions $[Q_0, \ldots, Q_n]$ given the image features $X$, where $Q_i$ is the $i^{th}$ question containing $m$ tokens $[Q_{i,0}, \ldots, Q_{i,m-1}]$. The Answer Generator predicts corresponding answers $[A_0, \ldots, A_n]$ given the image features and the questions. When generating $A_i$, attention masks are used to control the visibility of previous questions $[Q_0, \ldots, Q_{i-1}]$ and their answers $[A_0, \ldots, A_{i-1}]$, which we found are helpful contexts. During training, the ground-truth questions and answers are used as input (red arrows). During testing, model-predicted questions, optionally modified or extended by humans, are used as input (blue arrows), and their answers are concatenated to form the radiology report. Template questions on particular disease classes can be asked to extract numerical class probabilities. For simplicity, we omit details such as the input instructions in the figure.

respectively. As is shown in Figure 4, both modules utilize an identical transformer architecture: a MI2-based (Codella et al., 2024) visual backbone and a tiny text decoder of six transformer layers.

### 3.2.1 QUESTION GENERATION MODULE

The question generation module conducts sequence generation autoregressively with reference to the previously generated questions. Concretely, it generates $m$ output tokens $Q = (q_1, q_2, \ldots, q_m)$ by modeling Equation 1:

$$P(Q \mid X) = \prod_{i=0}^{m+1} P(q_i \mid X, q_0, q_1, \ldots, q_{i-1}), \tag{1}$$

The ground truth $Q$ is the concatenated questions. When providing inputs to the Answer Generator, we split $Q$ by the question mark ("?") to obtain individual questions.

### 3.2.2 ANSWER GENERATION MODULE

The Answer Generation learns to generate a sentence of $n$ tokens $Y_i = (y_{i_1}, y_{i_2}, \ldots, y_{i_n})$ conditioned on the image and a question $Q_i$. Mathematically, the module models the following:

$$P(Y_i \mid X, Q_i) = \prod_{j=0}^{n+1} P(y_{i_j} \mid X, Q_i, y_{i_0}, y_{i_1}, \ldots, y_{i_{j-1}}), \tag{2}$$

where $Y_i$ denotes the $i^{th}$ answer corresponding to question $Q_i$. By iterating $Q_i$ through all questions, the Answer Generator generates $n$ sentences $Y = (Y_1, Y_2, \ldots, Y_n)$ and composes the whole radiology report. In the interactive VQA mode, $Q_i$ is replaced by the tokenized user-entered question.

### 3.2.3 TRAINING RECIPE

**Training Stages**. The Question and Answer Generators are seperate modules using the same encoder-decoder architecture. The vision encoder is MedImageInsight (MI2) (Codella et al., 2024), a 0.36B-parameter model trained on medical images. The decoder is a six-layer, randomly initialized transformer text decoder of 0.07B parameters. The encoder and decoder are connected via a linear

projection layer. The total model size is $2 * (0.36 + 0.07) \approx 0.9$ B, around 13% of current state-of-the-art models that are based on 7B parameter models. We freeze the encoder and pre-train the decoder (the encoder is frozen) on CXR-697K, a pre-training dataset used in existing work (Chaves et al., 2024). Then, we duplicate the model and fine-tune for question and answer generation tasks, where the full model is made trainable.

**Mixture of VQA Data**. As discussed in subsection 3.1, our training data contain both report generation QA data and image classification QA to improve model performance. The data mixture ratio is discussed in Table 6.

**Prompt Templates**. Following existing studies, we use a short instruction which includes the Indication section when generating the questions and corresponding answers. The Indication section specifies the goal of the radiology study. We use the ground truth questions as input when training the Answer Generator.

**Attention Masks for Training Efficiency**. After converting the training data from image-report to image-QA pairs, the number of training samples increases by the number of sentences per report, which significantly increases training cost. To improve training efficiency, we concatenate all QA pairs for the same image and construct attention masks to control context visibility, thereby enabling us to run forward in one pass.

### 3.3 NUMERICAL CLASS CONFIDENCE EXTRACTION

#### 3.3.1 IMAGE CLASSIFICATION USING A TEXT GENERATION MODEL

$Q$Rad enables producing numerical class probabilities for medical findings, a capability absent in conventional report generation models. To extract these class probabilities, we leverage our VQA architecture by sending template classification questions to the model and request binary "yes" or "no" answers. We deliberately designed these responses to be single-token outputs, allowing us to extract clean probabilities directly from the model's output distribution, which evaluates the language model's intrinsic capability on distinguishing these classes. The probability for each class is computed using:

$$P(C_i = 1) = \frac{e^{x_\mathbf{yes}}}{e^{x_\mathbf{yes}} + e^{x_\mathbf{no}}}, \tag{3}$$

where $P(C_i = 1)$ represents the probability of the $i^{th}$ class, calculated from the softmax over $x_\mathbf{yes}$ and $x_\mathbf{no}$, the logits for [yes] and [no] being generated as the next token. Our approach is related to existing work (Kadavath et al., 2022) which uses $P($[true]$)$ as the confidence of an LLM in its answer. We use the softmax concerning both [yes] and [no] tokens to enable augmenting the text generation training data with image classification labels. This approach effectively transforms text generation over a binary vocabulary into a proxy for image classification, while sharing the same model weights with the report generation mode. In comparison, prior approaches represent binary classifications with free-text sentences that can span multiple tokens and display a high degree of stylistic variance, which makes extraction of clean class probabilities technically challenging.

### 3.4 FROM CLASS PROBABILITIES TO CALIBRATED CONFIDENCE SCORES

Theoretically, classifiers trained with proper scoring rules as the loss function naturally become calibrated (Blasiok et al., 2023; Fröhlich & Williamson, 2024). This applies to $Q$Rad, as the binary [yes]/[no] classification is trained with standard cross-entropy, a typical proper scoring rule. Recent work (e.g., ConfTuner (Li et al., 2025)) similarly uses single-token probabilities as confidence scores without calibration, which validates our design choice.

In reality, the extracted confidence scores may still benefit from post-hoc calibration due to challenges like class imbalance. In Table 1, we conducted calibration using temperature scaling, which improves the Expected Calibration Error (ECE), resulting in a better calibrated model. In addition, we provide an ROC evaluation in Appendix D.

Table 1: Expected Calibration Error (ECE) Before and After Calibration

| Classes | wAVG | Enl. | Car. | L.O. | L.L. | Ede. | Con. | Pmn. | Ate. | Pmt. | P.E. | P.O. | Fra. | S.D. |
|---|---|---|---|---|---|---|---|---|---|---|---|---|---|---|
| Ratio[1] | - | 0.62 | 0.54 | 0.63 | 0.13 | 0.44 | 0.38 | 0.22 | 0.45 | 0.07 | 0.38 | 0.17 | 0.31 | 0.47 |
| ECE (Before)[2] | 0.18 | 0.21 | 0.22 | 0.17 | 0.34 | 0.19 | 0.15 | 0.25 | 0.15 | 0.39 | 0.17 | 0.31 | 0.10 | 0.05 |
| ECE (After)[2] | 0.15 | 0.22 | 0.15 | 0.22 | 0.24 | 0.12 | 0.05 | 0.17 | 0.08 | 0.25 | 0.08 | 0.23 | 0.16 | 0.08 |

[1] **Ratio** is the percentage of positive samples, showing class imbalance in the MIMIC-CXR dataset. **wAVG** is the average of all classes weighted by their ratio

[2] **Before** and **After** show the ECE improvements from Temperature Scaling calibration

[3] The disease class shorthands represent Enlarged Cardiomediastinum, Cardiomegaly, Lung Opacity, Lung Lesion, Edema, Consolidation, Pneumonia, Atelectasis, Pneumothorax, Pleural Effusion, Pleural Other, Fracture, Support Devices, respectively

Table 2: Report Generation Performance on MIMIC-CXR

| Model | CheXbert | | | | | | | | RadGraph | BLEU | | ROUGE |
|---|---|---|---|---|---|---|---|---|---|---|---|---|
| | ("uncertain" as *negative*) | | | | ("uncertain" as *positive*) | | | | | | | |
| | Micro-avg | | Macro-avg | | Micro-avg | | Macro-avg | | | | | |
| | F1-14 | F1-5 | F1-14 | F1-5 | F1-14 | F1-5 | F1-14 | F1-5 | ER | (1) | (4) | (L) |
| *Single Image, Model size $\geq$ 7B* | | | | | | | | | | | | |
| LLaVA-Rad (Chaves et al., 2024) [F] | 57.3 | 57.4 | 39.5 | 47.7 | 57.3 | 60.2 | 44.0 | 53.3 | 29.4 | 38.1 | 15.4 | 30.6 |
| Med-Gemini (Yang et al., 2024) [F] | - | - | - | - | - | - | - | - | - | - | 20.5 | 28.3 |
| VILA-M3 40B (Nath et al., 2024) | - | - | - | - | - | - | - | - | - | - | **21.6** | 32.2 |
| Med-PaLM M (Tu et al., 2024) | 53.6 | 57.9 | 39.8 | **51.6** | - | - | - | - | - | 32.3 | 11.3 | 27.3 |
| MAIRA-1 (Hyland et al., 2023) [F] | 55.7 | 56.0 | 38.6 | 47.7 | 55.3 | 58.8 | 42.3 | 51.7 | 29.6 | **39.2** | 14.2 | 28.9 |
| GPT-4V | 35.5 | 25.8 | 20.4 | 19.6 | 35.6 | 33.3 | 25.3 | 29.6 | 13.2 | 16.4 | 1.9 | 13.2 |
| CheXagent (Chen et al., 2024) | 39.3 | 41.2 | 24.7 | 34.5 | 39.4 | 42.1 | 27.3 | 35.8 | 20.5 | 16.9 | 4.7 | 21.5 |
| LLaVA-Med (Li et al., 2024) [F] | 27.2 | 22.0 | 15.5 | 16.6 | 27.3 | 24.4 | 18.7 | 20.5 | 6.5 | 22.2 | 1.0 | 13.3 |
| LLaVA (Liu et al., 2024) [F] | 22.9 | 23.4 | 15.4 | 17.5 | 23.7 | 26.9 | 17.0 | 20.3 | 4.5 | 21.0 | 1.3 | 13.8 |
| *Q*Rad (ours, 4B) | **57.6** | **59.0** | **40.8** | 51.0 | **57.1** | **61.4** | **44.3** | **54.4** | 31.1 | 38.5 | 16.8 | **32.5** |
| *Single Image, Model size = 4B* | | | | | | | | | | | | |
| Baseline [b] | 54.3 | 55.2 | 36.9 | 46.6 | 54.1 | 57.4 | 40.4 | 50.5 | **31.1** | 40.1 | **17.8** | **32.7** |
| *Q*Rad (ours, 4B) [b] | **57.6** | **59.0** | **40.8** | **51.0** | **57.1** | **61.4** | **44.3** | **54.4** | **31.1** | **40.6** | 17.5 | 32.5 |
| *Single Image, Model size <1B* | | | | | | | | | | | | |
| PromptMRG (Jin et al., 2024) | - | - | - | - | - | - | - | - | - | - | 11.2 | 26.8 |
| Flamingo (Alayrac et al., 2022) | - | - | - | - | 51.9 | 56.5 | - | - | - | - | 10.1 | 29.7 |
| CvT2Dist. (Nicolson et al., 2023b) | 44.2 | - | 30.7 | - | - | - | - | - | - | 39.3 | 12.7 | 28.6 |
| $\mathcal{M}^2$ trans (Miura et al., 2020) | - | - | - | - | - | 56.7 | - | - | - | - | 11.4 | - |
| RGRG (Tanida et al., 2023a) | - | - | - | - | - | 54.7 | - | - | - | 37.3 | 12.6 | 26.4 |
| R2Gen (Chen et al., 2020b) | - | - | - | - | 22.8 | 34.6 | - | - | - | 35.3 | 10.3 | 27.7 |
| TieNet (Wang et al., 2018) | - | - | - | - | - | 27.1 | - | - | - | - | 8.1 | - |
| MI2 (Codella et al., 2024) | 56.3 | 57.9 | 38.4 | 49.3 | 55.7 | 59.3 | 43.2 | 52.1 | 28.5 | 37.3 | 15.3 | 31.7 |
| *Q*Rad [F] (ours, 0.9B) | **58.4** | **59.5** | **41.5** | **51.8** | **57.9** | **62.2** | **45.1** | **55.2** | **31.5** | **40.0** | 16.9 | **32.5** |

[F] The testing set includes only frontal-view images.

[a] The MAIRA-2 benchmark is redesigned to reflect clinical scenarios by combining multiple images from the same case into a single instance. Therefore, direct comparisons to other approaches cannot be made.

[b] The 4B models use BiomedCLIP (Zhang et al., 2023) as the vision encoder and Phi-3-mini (Abdin et al., 2024) as the text decoder.

Table 3: Performance on the ReXrank Benchmark

| Model | 1/RadCliQ | BLEU | BertScore | SembScore | RadGraph | RaTEScore | GREEN |
|---|---|---|---|---|---|---|---|
| UniRG-CXR[*] | 1.217 | 0.248 | 0.493 | 0.487 | 0.265 | 0.596 | 0.352 |
| *Q*Rad-0.9B, ours | 1.143 | 0.264 | 0.482 | 0.479 | 0.243 | 0.596 | 0.362 |
| MedVersa (Zhou et al., 2024) | 1.103 | 0.209 | 0.448 | 0.466 | 0.273 | 0.550 | 0.374 |
| Libra (Zhang et al., 2025a) | 0.898 | 0.232 | 0.402 | 0.403 | 0.218 | 0.523 | 0.356 |
| RadPhi3.5Vision (Ranjit et al., 2024) | 0.888 | 0.223 | 0.386 | 0.431 | 0.207 | 0.534 | 0.294 |
| CXRMate-ED (Nicolson et al., 2025) | 0.872 | 0.208 | 0.383 | 0.396 | 0.223 | 0.531 | 0.327 |
| CXRMate-RRG24 (Nicolson et al., 2024) | 0.870 | 0.198 | 0.367 | 0.423 | 0.220 | 0.521 | 0.338 |
| CheXpertPlus-CheX (Chambon et al., 2024) | 0.805 | 0.142 | 0.367 | 0.379 | 0.181 | 0.490 | 0.305 |
| DD-LLava-X[*] | 0.801 | 0.154 | 0.348 | 0.402 | 0.182 | 0.505 | 0.301 |
| RaDialog (Tanida et al., 2023b) | 0.799 | 0.127 | 0.363 | 0.387 | 0.172 | 0.485 | 0.273 |
| CheXpertPlus-MIMIC (Chambon et al., 2024) | 0.788 | 0.145 | 0.361 | 0.375 | 0.170 | 0.485 | 0.311 |
| RGRG (Tanida et al., 2023a) | 0.755 | 0.130 | 0.348 | 0.344 | 0.168 | 0.491 | 0.273 |
| MedGemma (Sellergren et al., 2025) | 0.744 | 0.165 | 0.346 | 0.339 | 0.159 | 0.549 | 0.293 |
| CheXagent (Chen et al., 2024) | 0.741 | 0.113 | 0.346 | 0.347 | 0.148 | 0.474 | 0.257 |
| MoERad-MIMIC[*] | 0.726 | 0.163 | 0.341 | 0.334 | 0.143 | 0.465 | 0.240 |
| Cvt2distilgpt2 (Nicolson et al., 2023a) | 0.719 | 0.126 | 0.331 | 0.329 | 0.149 | 0.432 | 0.268 |

[1] Results shown are for the Findings Generation task on the MIMIC-CXR dataset.
[2] Models are ranked by 1/RadCliQ-v1 (higher is better for all metrics). An introduction to metrics is available in Appendix A.
[*] UniRG-CXR, DD-LLava-X and MoERad from the leaderboard have no associated publications yet.

## 3.5 EXPERIMENTS AND ABLATION STUDIES

We conduct experiments on MIMIC-CXR (Johnson et al., 2023), one of the largest radiology report generation dataset. It has 227,835 image-report pairs. We use only the frontal view radiograph from each training sample. Following recent studies (Chaves et al., 2024; Hyland et al., 2023), we use the IU X-ray dataset (Demner-Fushman et al., 2016) as a fully held-out evaluation set. All 3198 frontal-view X-rays are used as the testing split unseen during training.

### 3.5.1 RADIOLOGY REPORT GENERATION

In Table 2 and Table 3, we evaluate our method on the official testing split of MIMIC-CXR. We provide both the conventional benchmark including lexical and clinical efficacy (CE) metrics and the newer ReXrank (Zhang et al., 2024) leaderboard. *Q*Rad outperforms major state-of-the-art methods across two benchmarks, despite using 13% the size of most existing models; Results on the IU X-ray dataset is available in Appendix E. Qualitative examples of generated questions and answers are included in Appendix C.

We include model training implementation details and introduction of evaluation metrics in Appendix A. The ROC-based evaluation per class is provided in Appendix D.

### 3.5.2 ABLATION STUDY AND HYPER-PARAMETERS

**Effectiveness of each component**: In Table 4, we conduct an ablation study on a MI2-based small model with 0.9B parameters. We first reframe the report generation task as a VQA process ("Caption-to-VQA"), and then augment the training data with image classification QA pairs ("Classification QA"). The table shows that each method brings consistent performance gains. The largest improvements are observed on Clinical Efficacy metrics (CheXbert, RadGraph), which reflect factual accuracy in the medical domain.

Appendix F compares implementation details of *Q*Rad across three dimensions, including the data mixture ratio, the source of pseudo-labels for the classification QA data and whether previous QA pairs are provided as input context. By comparing experiments (a) to (e), we found:

Table 4: Ablation Study on MIMIC-CXR

| Model | CheXbert | | | | | | | | RadGraph | BLEU | | ROUGE |
|---|---|---|---|---|---|---|---|---|---|---|---|---|
| | ("uncertain" as *negative*) | | | | ("uncertain" as *positive*) | | | | | | | |
| | Micro-avg | | Macro-avg | | Micro-avg | | Macro-avg | | | | | |
| | F1-14 | F1-5 | F1-14 | F1-5 | F1-14 | F1-5 | F1-14 | F1-5 | ER | (1) | (4) | (L) |
| **Baseline (MI2)** | | | | | | | | | | | | |
| *median* | 56.2 | 57.8 | 38.3 | 49.2 | 55.7 | 59.2 | 42.1 | 52.0 | 31.1 | 37.3 | 15.3 | 31.7 |
| *ci_l* | 55.1 | 56.2 | 36.7 | 47.1 | 54.7 | 57.8 | 40.6 | 50.6 | 30.5 | 36.8 | 14.9 | 31.2 |
| *ci_h* | 57.3 | 59.4 | 40.0 | 51.3 | 56.7 | 60.7 | 43.5 | 53.7 | 31.8 | 37.8 | 15.7 | 32.1 |
| **Baseline + Captioning-to-VQA** | | | | | | | | | | | | |
| *median* | 57.9 | 59.8 | 40.0 | 50.7 | 57.6 | 62.7 | 44.2 | 55.5 | 31.4 | 39.9 | 16.5 | 32.4 |
| *ci_l* | 56.8 | 58.3 | 38.1 | 48.9 | 56.6 | 61.3 | 42.6 | 53.9 | 30.8 | 39.3 | 16.0 | 31.8 |
| *ci_h* | 59.0 | 61.3 | 41.6 | 52.5 | 58.7 | 64.0 | 45.8 | 57.2 | 32.1 | 40.6 | 17.1 | 32.9 |
| **Baseline + Captioning-to-VQA + Classification QA (*Q*Rad)** | | | | | | | | | | | | |
| *median* | 58.3 | 59.5 | 41.5 | 51.8 | 57.9 | 62.2 | 45.1 | 55.2 | 31.6 | 40.2 | 16.7 | 32.5 |
| *ci_l* | 57.3 | 57.9 | 39.8 | 49.7 | 56.9 | 60.8 | 43.7 | 53.6 | 30.9 | 39.4 | 16.2 | 32.0 |
| *ci_h* | 59.4 | 61.0 | 42.97 | 53.7 | 59.0 | 63.5 | 46.6 | 57.0 | 32.2 | 40.9 | 17.2 | 33.1 |

[1.] The baseline ablates Captioning-to-VQA reframing, while keeping model architecture and pre-training the same. It is equivalent to the previous work in MedImageInsight (MI2) Codella et al. (2024).
[2.] To demonstrate statistical significance, we report the median and 95% confidence intervals (ci_l and ci_h) over 500 bootstrap replicates for all metrics.

- Performance is robust to data mixture ratios - (a) vs. (b)

- Using P+U as the positive label, which aligns with the "CheXbert: uncertain as positive" evaluation, leads to consistent performance gains across metrics. This is likely due to uncertain labels being corresponded to diseases mentioned in prior studies but ambiguously stated in current reports - (e) vs. (b), (c)

- Providing previous QA pairs as context improves performance - (d) vs. (b), (c), (e)

**Benefits of the Question Generator**: Appendix F - Table 7 demonstrates the importance of using a learned Question Generator over fixed template questions. The key challenge in medical report generation is the vast and complex space of possible medical conditions that can appear in an image. It is infeasible to enumerate all potential diseases as a predefined set of template questions. Moreover, even if such an exhaustive list existed, requiring the model to answer questions about every possible condition would be computationally prohibitive and inefficient. Our Question Generator addresses this by dynamically predicting relevant questions based on the input image, focusing only on conditions likely to be present.

**Quality of generated questions**: We observe that when given oracle questions that clearly specify each sentence's topic, the model shows substantial performance gains (Appendix F - Table 7). This demonstrates that stylistic variations (omissions, reordering) in the training data create noisy supervision signals, causing prior models to memorize surface patterns rather than learn medical content. Our model's strong performance with oracle questions proves it generates factually accurate answers. The differences between oracle and predicted questions represent legitimate stylistic choices rather than errors—these variations are natural in clinical practice.

## 4 CONCLUSION

In this paper, we introduce *Q*Rad, a novel approach that reframes long text generation from captioning to an Auto-VQA process. Our problem reformulation improves the factual quality, enables user interaction, and allows probability-based evaluation such as ROC curves. *Q*Rad improves the clinical utility of report generation with 13% of the model size.

## 5 ETHICAL CONSIDERATIONS

Medical datasets often contain sensitive patient information. To ensure the ethical use of such data, this study adheres to strict guidelines. All participants who accessed the MIMIC-CXR dataset, including the authors and radiologists involved in this research, completed the required onboarding process through PhysioNet[4]. For the IU X-ray dataset, we complied with the license[5].

To maintain compliance with PhysioNet's policy on the use of large language model APIs during the automatic evaluation, we utilized a secure, private, in-house deployment of GPT-4o. This approach guarantees that no sensitive information is shared with external parties.

Furthermore, to protect patient privacy, X-ray images presented in this paper were carefully selected from open, compliance-free sources, ensuring that no identifiable patient information is disclosed.

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

## A   IMPLEMENTATION DETAILS AND EVALUATION METRICS

**Training Parameters**. We use an image size of 512x512. During the pre-training on CXR-697K (consistent with existing work LLaVA-Rad (Chaves et al., 2024)), we freeze the image encoder and updates the text decoder with a learning rate of 2E-5 for 400 epochs. The batch size is 2048, and no instruction is used in this phase. When fine-tuning the model on MIMIC-CXR VQA, we use a learning rate of 1E-5 for the image encoder and 5E-5 for the text decoders. The training takes 5 hours with 128 V100 GPUs, using a batch size is 512 and set for 60 epochs. We mixed the report generation QA and image classification QA by a ratio of 6:4.

**Evaluation Metrics**. Table 2: CheXbert (Smit et al., 2020) is a Clinical Efficacy (CE) metric that classifies generated reports into 14 disease categories and evaluates the F1 scores, focusing on factual accuracy rather than textual overlap. As CheXbert produces an *uncertain* class in additional to positive and negative classes, existing methods take *uncertain* as either positive and negative to evaluate. RadGraph-ER (Jain et al., 2021a) is designed specifically for radiology reports and assesses the correctness of extracted entities and their attributes. BLEU and ROUGE are standard lexical metrics that measure n-gram similarity to evaluate text overlap. We use results from (Chaves et al., 2024) if not available in the original papers. Table 3: ReXrank (Zhang et al., 2024) is a newer proposed benchmark for radiology report generation. The leaderboard ranks models by the inverse of RadCliQ (Yu et al., 2023), a composite metric combining BLEU-2 (Papineni et al., 2002), BertScore (Zhang et al., 2019), SembScore (Smit et al., 2020), and RadGraph-F1 (Yu et al., 2023), where BertScore and SembScore are embedding similarities from Bert and CheXbert, respectively. Other individual metrics reported include RaTEScore (Zhao et al., 2024) and GREEN Ostmeier et al. (2024), where RaTEScore is based on embedding similarities of extracted medical entities, and GREEN is an LLM-based metric.

## B   ADDITIONAL COMPARISON WITH RELATED WORK

**Structured Report Generation**: although free-form radiology reports offer flexibility in clinical use, they introduce challenges for generation and evaluation due to linguistic variability. Structured Report Generation (Delbrouck et al., 2025) was proposed as a new task that standardizes report formats by organizing content under fixed topics (e.g., lungs, airways, pleura). Rad-ReStruct (Pellegrini et al., 2023a) further casts each topic as single- or multi-label classification to enable F1-based evaluation. In contrast, $Q$Rad produces free-text reports, where the topic of each sentence are not fixed, but are generated dynamically from the input.

**General-domain LLMs**: to enable text-only LLMs (e.g., GPT-3) to perform VQA, existing work (Özdemir & Akagündüz, 2024; Zhang et al., 2025b; Hu et al., 2022; Serra et al., 2025) usessss image captioning models to describe the image for the LLM. $Q$Rad proposes using captioning-to-VQA reframing to improve image captioning, which is a different task from these methods. Besides, the Auto-VQA part of our work differs from conventional VQA in that our model learns to predict both the questions and the answers, where the chain of questions specifies the structure of the output text.

Regarding the interactive capability, RaDialog (Pellegrini et al., 2023b) fits a conversational VLM from the general domain for radiology report generation. Compared to our work, the LLMs and VLMs from the general domain inherit stronger conversational capabilities, but there are significant performance gaps in clinical metrics compared to $Q$Rad. In the ReXrank leaderboard, $Q$Rad outperforms RaDialog by 43.1% in the composite metric.

## C   QUALITATIVE RESULTS

Qualitative examples illustrating the $Q$Rad pipeline are shown in Figure 5. The figure highlights three key aspects:

- **Intrinsic VQA capability:** The predicted answers are directly relevant to the input questions, demonstrating the model's ability to perform visual question answering.

- **Factual correctness:** The model generates factually accurate answers, although there may be stylistic differences such as sentence structure or order.

- **Interactive refinement:** When provided with ground-truth questions (simulating a scenario where a radiologist requests specific information), the model produces answers that are both reasonable and closely aligned with the ground-truth responses.

## D ROC-BASED EVALUATION FOR REGULATORY VALIDATION

*Q*Rad is the first report generation model to produce class probabilities scores for defined disease directly from its text generation components. Unlike multi-task models that use separate modules for classification and text generation, *Q*Rad generates both outputs from the same component. This design enables the evaluation of confidence scores to directly reflect the model's intrinsic classification capability.

The class probabilities are used to generate Receiver Operating Characteristic (ROC) and sensitivity-specificity curves, which are typical in FDA approval studies for diagnostic systems. As shown in Figure 6, this confidence-based evaluation provides more granular insights into clinical utility, such as the characteristics of the sensitivity and specificity trade-off. This is especially valuable because clinical applications often have different costs for false positives and false negatives.

From Figure 6, we observe that *Q*Rad performs reliably on classes such as Enlarged Cardiomediastinum, Cardiomegaly, and Lung Opacity, but is less reliable on Pleural Other, Fracture, and Pneumothorax. We attribute this difference to two main factors. First, conditions like fracture require detection of subtle details and are rare in the dataset. Second, some classes (e.g., Pleural Other) aggregate many rare disease names, making it challenging for our prompts to comprehensively elicit the expected output.

The ROC curve enables more comprehensive guidance for clinical adoption by illustrating the model's characteristics across different sensitivity-specificity operating points, rather than relying solely on binary predictions as in existing CheXbert-based metrics. For example, in a copilot system that alerts radiologists to potential missed findings, maximizing sensitivity may be prioritized to ensure that as few true cases as possible are overlooked. Conversely, in automated triage systems that escalate only the most critical or certain cases for urgent review, higher specificity may be preferred to avoid unnecessary interruptions and reduce alarm fatigue. The ROC curve allows stakeholders to evaluate the model's behavior on disease classes relevant to the clinical context and risk tolerance, thereby assessing its practical utility more faithfully.

## E RESULTS ON IU X-RAY

Following recent studies Chaves et al. (2024); Bannur et al. (2024), we use the IU X-ray dataset Demner-Fushman et al. (2016) as a fully held-out evaluation set. All 3198 frontal-view X-rays are used as the testing split unseen during training. Results in Table 5 shows that *Q*Rad generates well on unseen data.

## F ADDITIONAL TABLES FOR ABLATION STUDY AND HYPER-PARAMETERS

We provide additional tables for ablation studies and hyper-parameters such as dataset mixture ratios in Table 6 and Table 7.

Table 5: Report Generation Performance on IU-XRay

| Model | CheXbert ("uncertain" as *negative*) | | | | CheXbert ("uncertain" as *positive*) | | | | RadGraph | BLEU | | ROUGE |
|---|---|---|---|---|---|---|---|---|---|---|---|---|
| | Micro-avg | | Macro-avg | | Micro-avg | | Macro-avg | | | | | |
| | F1-14 | F1-5 | F1-14 | F1-5 | F1-14 | F1-5 | F1-14 | F1-5 | ER | (1) | (4) | (L) |
| R2Gen Chen et al. (2020b) | - | - | 13.6 | - | - | - | - | - | - | 32.5 | 5.9 | 25.3 |
| CvT2Dist. Nicolson et al. (2023b) | - | - | 16.8 | - | - | - | - | - | - | 38.3 | 8.2 | **27.7** |
| RGRG Tanida et al. (2023a) | - | - | 18.0 | - | - | - | - | - | - | 26.6 | 6.3 | 18.0 |
| LLaVA-Rad Chaves et al. (2024) | **53.5** | - | - | - | - | - | - | - | - | - | - | 25.3 |
| *Q*Rad | 46.5 | 36.9 | **27.0** | 27.2 | 44.3 | 38.8 | 28.7 | 31.2 | 29.4 | **41.9** | **10.8** | 25.3 |

Table 6: Comparison of Dataset Hyper-parameters on MIMIC-CXR

| Model | CheXbert ("uncertain" as *negative*) | | | | CheXbert ("uncertain" as *positive*) | | | | RadGraph | BLEU | | ROUGE |
|---|---|---|---|---|---|---|---|---|---|---|---|---|
| | Micro-avg | | Macro-avg | | Micro-avg | | Macro-avg | | | | | |
| | F1-14 | F1-5 | F1-14 | F1-5 | F1-14 | F1-5 | F1-14 | F1-5 | ER | (1) | (4) | (L) |
| **(a) Classification QA = 20%, Label={P}** | | | | | | | | | | | | |
| *median* | 57.9 | 59.6 | 40.4 | 51.2 | 57.4 | 61.8 | 44.6 | 54.7 | 31.4 | 40.5 | 16.6 | 32.4 |
| *ci_l* | 56.8 | 57.8 | 38.8 | 49.1 | 56.4 | 60.5 | 43.1 | 53.2 | 30.7 | 39.8 | 16.1 | 31.8 |
| *ci_h* | 59.0 | 61.1 | 42.0 | 53.1 | 58.5 | 63.3 | 46.2 | 56.4 | 32.0 | 41.2 | 17.2 | 32.9 |
| **(b) Classification QA = 40%, Label={P}** | | | | | | | | | | | | |
| *median* | 57.8 | 59.5 | 40.2 | 50.9 | 57.3 | 61.7 | 44.5 | 54.4 | 31.4 | 40.0 | 16.6 | 32.5 |
| *ci_l* | 56.7 | 58.0 | 38.6 | 48.9 | 56.4 | 60.2 | 43.0 | 52.9 | 30.8 | 39.3 | 16.1 | 31.9 |
| *ci_h* | 58.9 | 61.0 | 41.9 | 52.7 | 58.4 | 63.1 | 46.2 | 56.2 | 32.1 | 40.8 | 17.1 | 33.1 |
| **(c) Classification QA = 40%, Label={P, Random U}** | | | | | | | | | | | | |
| *median* | 57.9 | 59.2 | 40.5 | 50.8 | 57.5 | 61.6 | 44.5 | 54.5 | 31.3 | 40.1 | 16.6 | 32.4 |
| *ci_l* | 57.0 | 57.7 | 38.8 | 48.8 | 56.5 | 60.3 | 42.9 | 53.0 | 30.7 | 39.4 | 16.1 | 31.9 |
| *ci_h* | 59.1 | 60.7 | 42.0 | 52.6 | 58.5 | 63.1 | 46.0 | 56.3 | 31.9 | 40.8 | 17.2 | 33.0 |
| **(d) Classification QA = 40%, Label={P, U}, w/o QA Context** | | | | | | | | | | | | |
| *median* | 56.6 | 58.8 | 39.6 | 50.7 | 56.3 | 61.3 | 43.4 | 54.1 | 28.0 | 41.8 | 13.8 | 27.9 |
| *ci_l* | 55.5 | 57.4 | 38.2 | 48.7 | 55.4 | 59.9 | 42.1 | 52.6 | 27.5 | 41.3 | 13.5 | 27.5 |
| *ci_h* | 58.0 | 60.3 | 41.1 | 52.5 | 57.4 | 62.7 | 44.8 | 55.8 | 28.6 | 42.3 | 14.2 | 28.3 |
| **(e) Classification QA = 40%, Label={P, U}** | | | | | | | | | | | | |
| *median* | 58.3 | 59.5 | 41.5 | 51.8 | 57.9 | 62.2 | 45.1 | 55.2 | 31.6 | 40.2 | 16.7 | 32.5 |
| *ci_l* | 57.3 | 57.9 | 39.8 | 49.7 | 56.9 | 60.8 | 43.7 | 53.6 | 30.9 | 39.4 | 16.2 | 32.0 |
| *ci_h* | 59.4 | 61.0 | 42.97 | 53.7 | 59.0 | 63.5 | 46.6 | 57.0 | 32.2 | 40.9 | 17.2 | 33.1 |

[1.] The Classification QA ratio (0%, 20%, 40%) indicates the proportion of classification QA pairs in the training data.
[2.] The Label field defines which CheXbert classes are mapped to the positive classes in the Classification QA data: P (positive only), U (positive and uncertain), Random U (uncertain randomly used as positive)
[3.] QA Context represents that whether previous QA pairs are provided as input context. The w/o QA Context ablates this feature.

(a) *Q*Rad Predicted Questions-Answer Pairs

- [Q1] "What is the position of the endotracheal tube?"
  [A1] "The endotracheal tube is 3.5 cm above the carina."

- [Q2] "Where does the right-sided picc line terminate?"
  [A2] "A right-sided picc line terminates in the mid svc."

- [Q3] "What is the course of the enteric tube?"
  [A3] "An enteric tube courses below the diaphragm with the tip out of the field of view."

- [Q4] "Are there any changes in the lung opacities compared to previous studies?"
  [A4] "Since the prior study, there has been improvement in the bilateral parenchymal opacities."

- [Q5] "What is the status of the lung volumes and the cardiomediastinal silhouette?"
  [A5] "Lung volumes are low and the cardiomediastinal silhouette is enlarged, but stable."

(b) *Q*Rad Concatenated Report

- [Report] "The endotracheal tube is 3.5 cm above the carina.  A right-sided picc line terminates in the mid svc.  An enteric tube courses below the diaphragm with the tip out of the field of view.  Since the prior study, there has been improvement in the bilateral parenchymal opacities. Lung volumes are low and the cardiomediastinal silhouette is enlarged, but stable."

(c) *Q*Rad Prediction Compared with Ground-truth

- [GT-Q1] "Are there any support devices present?"
  [GT-A1] "An endotracheal tube, NG tube, and right upper extremity PICC with its tip at the cavoatrial junction are unchanged."
  [PD-A1] "An endotracheal tube is in place with the tip terminating 3.5 cm above the carina.  A nasogastric tube is seen coursing below the diaphragm with the tip terminating in the left upper quadrant.  A right picc is unchanged in position with the tip terminating in the mid-to-low svc."

- [GT-Q2] "Is there any change in lung opacity?"
  [GT-A2] "There is no change in left lower lobe opacity."
  [PD-A2] "There has been interval improvement in the right basilar opacity."

- [GT-Q3] "Are there signs of pleural effusion or pneumothorax?"
  [GT-A3] "There is no large pleural effusion, or pneumothorax."
  [PD-A3] "No pleural effusion or pneumothorax is detected."

- [GT-Q4] "What is the condition of the cardiac silhouette and mediastinal contours?"
  [GT-A4] "The cardiac silhouette remains moderately enlarged, mediastinal contours are notable for calcification of the aortic arch."
  [PD-A4] "The cardiac silhouette is mildly enlarged but stable, the mediastinal contours are prominent but stable."

Figure 5: Qualitative example of question-answer generation by *Q*Rad. (a) displays a set of questions ($Q_n$) generated by *Q*Rad for a given image, along with their corresponding answers ($A_n$). (b) demonstrates how the answers are concatenated to form the complete radiology report. (c) compares ground-truth questions [GT-$Q_n$], ground-truth answers [GT-$A_n$] from the reference report, and *Q*Rad's predicted answers [PD-$A_n$] for each question. *Q*Rad demonstrates factually reliable outputs, even if the order of information differs.

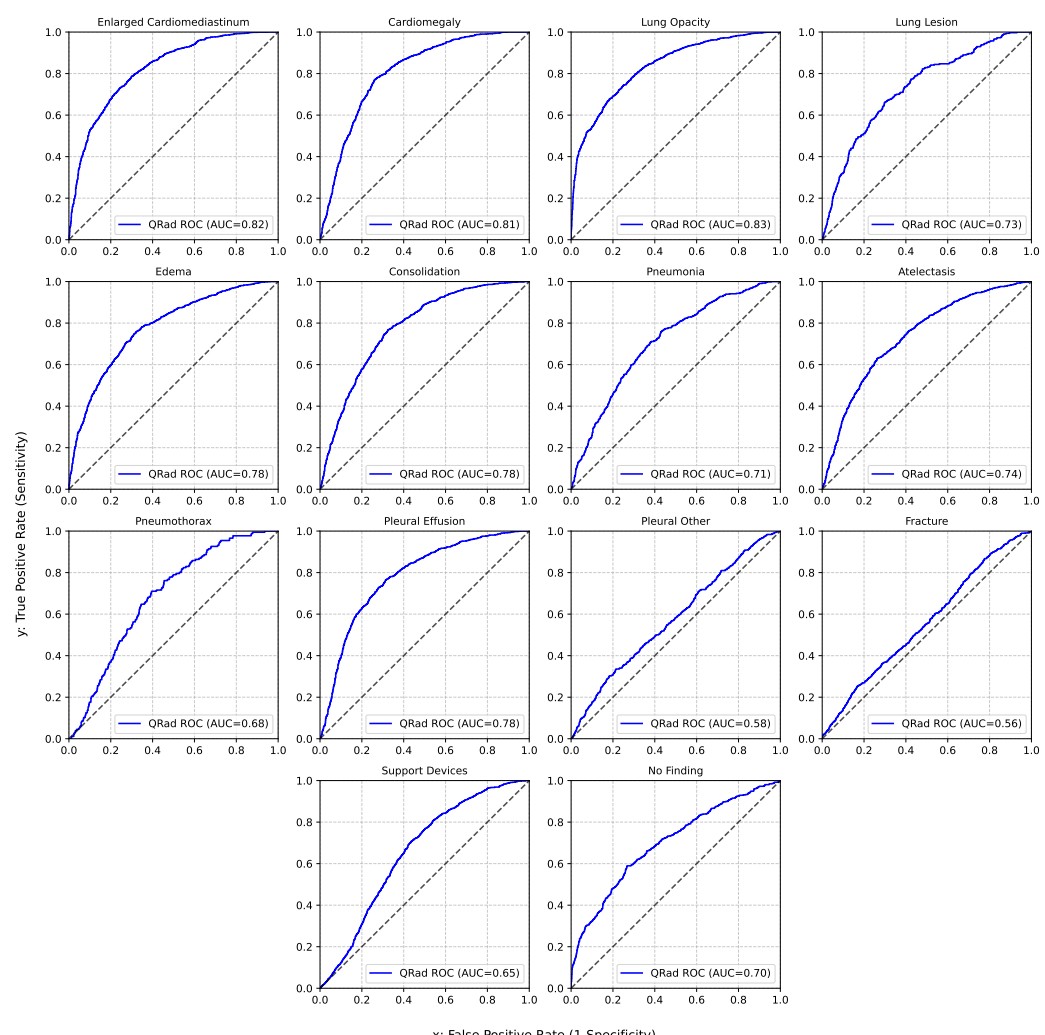

Figure 6: Receiver Operating Characteristic (ROC) curves for $Q$Rad across multiple disease classes. The x-axis shows the False Positive Rate (1-specificity), and the y-axis shows the True Positive Rate (sensitivity). Each curve illustrates the trade-off between sensitivity and specificity. The ROC analysis enables a nuanced assessment of $Q$Rad's clinical utility across different operating points and disease categories.

Table 7: Ablation of the Question Generator

| Model | CheXbert | | | | | | | | RadGraph | BLEU | | ROUGE |
|---|---|---|---|---|---|---|---|---|---|---|---|---|
| | ("uncertain" as *negative*) | | | | ("uncertain" as *positive*) | | | | | | | |
| | Micro-avg | | Macro-avg | | Micro-avg | | Macro-avg | | | | | |
| | F1-14 | F1-5 | F1-14 | F1-5 | F1-14 | F1-5 | F1-14 | F1-5 | ER | (1) | (4) | (L) |
| Template Questions [a] | 44.7 | 52.1 | 28.5 | 37.5 | 46.6 | 56.3 | 34.2 | 45.5 | 23.5 | 31.1 | 8.2 | 20.9 |
| Predicted Questions [b] | **58.4** | **59.5** | **41.5** | **51.8** | **57.9** | **62.2** | **45.1** | **55.2** | **31.5** | **40.0** | **16.9** | **32.5** |
| Oracle Questions [c] | 74.7 | 78.0 | 60.5 | 72.2 | 76.1 | 79.8 | 66.0 | 74.9 | 48.0 | 54.4 | 30.5 | 52.8 |

[a] We use the template questions for all input images composed from the 14 widely used CheXbert classes
[b] We use the Question Generator to learn and predict the questions per input image (the $Q$Rad method)
[c] Oracle questions are ChatGPT-generated directly from the ground truth reports. This is the ground truth used to train the Question Generator

