# OpenReview forum: "QRad: Enhancing Radiology Report Generation by Captioning-to-VQA Reframing"
_ICLR.cc/2026/Conference — Submitted to ICLR 2026_

### Official Review · Reviewer_xLL4 · 2025-10-20

**Soundness:** 4
**Presentation:** 3
**Contribution:** 3
**Rating:** 6
**Confidence:** 5

**Summary:**

They frame radiology report generation as a QA process, where the final report consists of a set of chained answers. For training, QA pairs are generated by splitting the report into a set of answers and letting GPT-4 generate a corresponding question for each answer. The results show modest improvements with the proposed method.

**Strengths:**

- Formulating free-text report generation as VQA task is novel and targets very relevant problems in report generation, such as ambiguous validation and linguistic ambiguity.
- The architecture is simple but efficient and effective.
- The formulation allows for an interactive mode additionally to just writing the report.
- Ablation study for validating the effectiveness of different components
- Clear reporting of all variants of CheXbert score
- Extensive comparison to prior work

**Weaknesses:**

- I am missing an architecture / flow figure showing what is the input, what is generated from that, how does the final output look etc.
- The language modeling loss during training relies on teacher forcing, so actually C,A,B is not penalized a lot compared to A,B,C if the sentences describing each of these findings remain correct as only the very beginning of the sentence is penalized. The formulations within A,B,C are penalized the same in Q-Rad given that for the training of answer generation again language modeling loss is used. The motivation section should be adapted to be technically accurate about this topic.
- A ROC curve is not the same as a confidence score, the third paragraph in the introduction completely mixes these things up.
- The concepts proposed in this paper are related to structured report generation, which should therefore be discussed in the related work section.
- RGRG and RaDialog should be discussed as a related works using a two-step approach. RGRG proposes a 2-level approach of first selecting abnormal regions and then specifying the findings, and RaDialog first classifies findings on a high level (using the same labels as the proposed image classification QA) before generating a final report.

**Questions:**

- How can you assure that these two artifacts are actually aligned, so that the scores from the binary answers can be used for judging the quality of the generated report?
- Why are many models from Table 2 excluded in Table 1, even if they were evaluated on MIMIC-CXR report generation?

---

> ### Author Response · Authors · 2025-12-03
>
> ### W1: Missing flowchart
>
> Thank you for the constructive suggestions. We have added a flowchart with details in the revision (Table 1).
>
> ---
>
> ### W2: Technical accuracy on teacher forcing
>
> We appreciate you pointing this out. We acknowledge the lack of technical accuracy when we tried to illustrate the idea intuitively. We have revised the related part for better technical accuracy.
>
> ---
>
> ### W3: Mixing up confidence score and probability
>
> This is a very constructive feedback – thank you for helping us improve our paper. We have revised our writing for better accuracy. Besides, we add the contents in Page 7 to better contextualize our work of using the token probability as confidence. In Page 8, we added the calibration and ECE evaluation.
>
> ---
>
> ### W4: Relationship to structured report generation
>
> This is a great suggestion. We have added this content to the related work. Thank you for your input.
> Briefly, structured report generation requires the same format for all reports to help generation and evaluation. We aim at generating free-text reports which are widely adopted in clinical scenarios. Our generated questions captures the structure of the report dynamically based on the input image.
>
> ---
>
> ### W5: Compare with RGRG, RaDialog as related two-step approach.
>
> Thanks for the suggestions. We have added them in our related work and appendix.
>
> ---
>
> ### Q1: The alignment of class probability and the textual report
>
> Thanks for this great question.
> The classification ground truth is derived from the ground truth report, which we expect them to reasonably align with each other. However, in reality, we acknowledge the possibility of confliction between two outputs. We think this is primarily a threshold issue (the textual report is binary, and the class probabilities are continuous and numerical). We think a possible solution is to post process the report using an LLM as a form of self-alignment or self-correction.
>
> ---
>
> ### Q2: Different models in Table 1 and Table 2.
>
> Thank you for noticing the difference. Table 1 (Table 2 in the revision) are conventional metrics that earlier work use. Table 2 (Table 3 in the revision) is the ReXrank leaderboard, a new benchmark which became popular this year. Some new models only evaluates on ReXRank, causing mismatch. As ReXrank is a live benchmark, we will sync both tables as much as possible in the final version of the paper.

---

### Official Review · Reviewer_qxPY · 2025-10-20

**Soundness:** 2
**Presentation:** 2
**Contribution:** 2
**Rating:** 2
**Confidence:** 3

**Summary:**

The paper proposes QRad, a novel framework that reframes radiology report generation from a traditional image captioning problem into a self-directed visual question answering (VQA) process. The model first generates a chain of questions relevant to the X-ray image (capturing the structure of the report) and then answers each question, concatenating the results into a coherent report. This reformulation mitigates overfitting to linguistic variations, improves factual accuracy, supports interactive querying, and allows for ROC-based evaluation—a step toward clinical validation.

**Strengths:**

1.  Comparing with the tradition method to generate the radiology report, it has much improvement.

**Weaknesses:**

1. The contribution of this work appears limited in scope for the broader research community. It does not provide substantial technical or theoretical advancements in the field of machine learning or deep learning. The study mainly builds upon previous question–answering research [1,2,3,4] and extends it by incorporating image inputs.

2. The paper claims to present "a novel approach that reframes long text generation"; however, the generation results appear to be highly dependent on the underlying LLM, which raises concerns about the novelty and contribution of the proposed method.

3. In Table 2, the reported improvement is not significant, which raises questions about the effectiveness and robustness of the proposed approach.

4. Limited Generalization Beyond X-rays Although the paper claims extensibility to other domains, all experiments are confined to chest X-rays (MIMIC-CXR, IU-Xray). There is no evidence that the approach generalizes to CT, MRI, or multi-view imaging, which are common in clinical workflows.

[1] Enhancing Visual Question Answering through Question-Driven Image Captions as Prompts

[2] SCRA-VQA: Summarized Caption-Rerank for Augmented Large Language Models in Visual Question Answering

[3] PromptCap: Prompt-Guided Task-Aware Image Captioning

[4] All You May Need for VQA are Image Captions

**Questions:**

see weakness

---

> ### Author Response · Authors · 2025-12-03
>
> ***We appreciate the effort in reviewing our paper. We believe the related work are substantially different from our task and our method.***
>
> ---
>
> ### W1: Concern on the contribution
>
> The listed papers [1-4] are technically disconnected from our paper. [1–3] enable text-only LLMs (e.g., GPT-3) to perform VQA, where they use image captioning models to describe the image for the LLM. [4] is an early VQA data generation method predating GPT-3.5, aimed at constructing QA datasets rather than improving captioning models.
>
> In contrast, our work focuses on **enhancing the image captioning task** (report generation) . In our proposed reframing method, we train the model to predict both the questions and the answers, which is also different from the conventional VQA.
>
> ---
>
> ### W2: highly dependent on the underlying LLM
>
> The LLM is used in offline data processing, and is not the focus of our paper. Splitting the reports into sentences and writing corresponding questions are straight-forward tasks that most contemporary LLMs handle reliably.
>
> Below, we use three LLMs to process the data (using the same prompt), and compare the final performance. While LLMs with stronger instruction following capabilities yield better supervision, QRad variants trained with these supervisions still achieve the same ranking on the ReXrank leaderboard.
>
> The core contribution of our paper is not data processing but the **reformulation** from text generation to Auto-VQA, where the model predicts both the questions and the answers. This framework achieves significant performance gain, and is independent of any specific upstream or model architecture.
>
> | ReXrank Ranking | Model                                   | 1/RadCliQ-v1 | BLEU  | BertScore | SembScore | RadGraph | RaTEScore |
> |---------|-------------------------------------------|--------------|-------|-----------|-----------|----------|-----------|
> | 1       | UniRG-CXR 7B                           | 1.217        | 0.248 | 0.493     | 0.487     | 0.265    | 0.596     |
> | 2       | QRad 0.9B (GPT-4o)                        | 1.143        | 0.264 | 0.482     | 0.479     | 0.243    | 0.596     |
> |         | QRad 0.9B (grok-4-fast-non-reasoning)     | 1.112        | 0.267 | 0.477     | 0.467     | 0.239    | 0.591     |
> |         | QRad 0.9B (gpt-5)                         | 1.106        | 0.261 | 0.476     | 0.470     | 0.235    | 0.589     |
> | 3       | MedVersa 7B                               | 1.103        | 0.209 | 0.448     | 0.466     | 0.273    | 0.550     |
>
> ---
>
> ### W3: Improvement is not significant, raising questions about the effectiveness and robustness of the proposed approach
>
> Thank you for recognizing that our method has much improvement in the _Strengths_. We hope the interpretation below can help illustrate the significance and robustness of the improvement:
>
> - In Table 2 (Table 3 in the revision), QRad achieves the second place with only 13% the model parameters of existing work. Specifically, QRad has 0.9B parameters while existing work mostly have 7B or even larger architectures. The first-ranking model, UniRG-CXR (no paper associated yet) applies RFT with 1/RadCliQ as the reward, and our method is orthogonal to it.
> - Table 1 (Table 2 in the revision) shows detailed Clinical Efficacy metrics where QRad outperforms prior larger in most columns. We reproduced results on the same level using a different encoder and decoder, showing the robustness of the proposed reframing method to the implementation. For statistical significance, we follow existing work to report the median and 95% confidence intervals in Table 3. The median value still outperforms prior studies, showing the robustness of our approach.
>
> ---
>
> ### W4: Limited Generalization Beyond X-rays
>
> Thank you for the suggestion. As shown in our paper, our captioning-to-VQA reframing is agnostic to the underlying encoder/decoder, making it extendable to various domains. We follow the majority of existing work [1] in evaluating on the CXR task due to data availability. We will remove any claims on CT and MRI in our revision.
>
> About multi-view CXR:
>
> The ReXRank leaderboard covers both single-image and multi-image methods. QRad ranks second on ReXrank (Table 3), ahead of multi-image methods; the top model (UniRG-CXR) also uses a single frontal view.
>
> Prior work shows that multi-view inputs offer limited benefit for CXR report generation, which motivates our choice of a single image for efficiency. Architecturally, QRad can accept multiple images in the same manner as encoder–decoder models—by concatenating visual tokens before the decoder.
>
> [1] https://github.com/mk-runner/Awesome-Radiology-Report-Generation
>
> ---
>
> ***We hope our clarifications on the existing work and our performance gain can address your concerns.***

---

### Official Review · Reviewer_E7Na · 2025-10-26

**Soundness:** 3
**Presentation:** 2
**Contribution:** 2
**Rating:** 4
**Confidence:** 4

**Summary:**

The paper introduces a novel method called QRad, which aims to reframe the task of radiology report generation. While traditional approaches typically follow a direct image-to-text path (i.e., image captioning), QRad adopts a two-stage, self-directed Visual Question-Answering (VQA) process.

This process comprises two core modules:

1. A Question Generation Module ($f_Q$) that generates a chain of clinically relevant questions based on the input radiographic image.

2. An Answer Generation Module ($f_A$) that provides answers to these questions.

The final radiology report is constructed by sequentially concatenating these answers.

The authors claim the method offers three major advantages: (1) It improves the report's factual accuracy by reducing overfitting to linguistic style variations.(2) The model has an intrinsic interactive capability, allowing clinicians to ask follow-up questions to obtain more information.(3) It is capable of extracting numerical confidence scores to support Receiver Operating Characteristic (ROC) curve-based evaluation, which is critical for regulatory approval.

**Strengths:**

This paper shows strong potential across four key areas: clinical relevance, evaluation, design, and efficiency.

- It tackles the critical clinical need for automated, reliable radiology report generation. This addresses the goal of reducing radiologists' workload and improving diagnostic efficiency, promising a significant clinical impact.

- The paper proposes a novel and practical evaluation methodology. Extracting numerical confidence scores via template-based VQA is a major contribution. This solves the generative models' critical bottleneck—the lack of quantifiable confidence—which is essential for regulatory validation (like ROC analysis for FDA approval). This provides a viable path for deployment beyond typical NLP metrics.

- The design features conceptual elegance. The idea of decomposing the task—separating factual content generation (answers) from report structure planning (questions)—is ingenious. It offers a principled solution to the "loss-metric mismatch" problem, allowing the model to focus purely on factual accuracy instead of stylistic variations.

- The model demonstrates impressive efficiency and performance. QRad, with a much smaller size (0.9B parameters), outperforms larger SOTA models (often >7B parameters) on both linguistic and clinical metrics. This high parameter efficiency is highly valuable for real-world applications and deployment.

**Weaknesses:**

Despite its merits, the paper presents several significant deficiencies that impact the evaluation of its contribution, making it difficult to meet the acceptance standards of a top-tier conference.

- **Major Concerns Regarding Novelty and Literature Context:** The paper's claim of core innovation—reframing report generation as a VQA process—is unsubstantiated. Multiple prior works have already explored VQA-based report generation[1] and interactive report generation[2]. Furthermore, the paper's two-stage "Question-Answer" process is conceptually similar to the RG-AG pipeline[3]. QRad's failure to acknowledge these precedents presents its contribution as a paradigm shift rather than the more accurate incremental contribution (e.g., from structured VQA to free-text VQA). The authors must clearly define their specific, narrower contribution.

- **Insufficient Methodological Elaboration:** The paper lacks a detailed flowchart to explain the data flow and architectural specifics of the two modules, particularly regarding weight sharing and training strategies when both modules use the same Transformer architecture. Given that the core concept is not entirely new, the method's technical contribution appears to be incremental, with innovation stemming primarily from data preparation rather than the model architecture itself.

- **Lack of Qualitative and Clinical Validation:** A critical omission is the absence of a complete generated report example. The paper only shows answers to individual questions but never the final report constructed by concatenating these answers. This simple concatenation may introduce problems with coherence, repetition, and logical flow, making it impossible to evaluate the report's true clinical utility and readability. Evaluating component-level accuracy is an insufficient substitute for assessing the quality of the final, integrated product (i.e., the complete report).

[1] Rad-ReStruct: A Novel VQA Benchmark and Method for Structured Radiology Reporting

[2] RaDialog: A Large Vision-Language Model for Radiology Report Generation and Conversational Assistance

[3] Grounding Chest X-Ray Visual Question Answering with Generated Radiology Reports

**Questions:**

Based on the analysis above, the authors are requested to clarify the following points:

The paper’s core innovation claim is reframing report generation as a VQA process. However, prior work such as Rad-ReStruct [1] already modeled structured reporting as a hierarchical VQA task. Given this, could the authors explicitly articulate the novelty of their contribution relative to these existing VQA-based reporting frameworks and specifically delineate how their approach differs?

The paper presents interactivity as a key advantage. The RaDialog[2] is an end-to-end conversational VLM specifically designed for interactive report generation and correction. How does QRad’s emergent interactivity compare to the specifically engineered dialogue capabilities of models like RaDialog? What are the relative pros and cons of the two methods?

The two-stage, Chain-of-Thought-inspired decomposition is central to the method. The RG-AG pipeline [3] also proposes a similar two-stage decomposition, but uses a generated draft report as the intermediate context, rather than questions. Could the authors comment on this alternative and justify the choice of using "questions" as the intermediate "thought" artifact, as opposed to other potential textual representations (such as a draft report)?

The methodological description would be significantly improved by a more detailed architectural diagram. Could the authors provide a flowchart that clearly details the precise data flow, including tensor dimensions, the interaction between the shared visual encoder and the two separate decoder modules, and any attention masking mechanisms used for efficient training?

A critical piece of evidence is missing: the final, end-to-end generated report. Could the authors provide several examples of complete reports (stitched together from the answers), covering both normal and abnormal cases, juxtaposed with ground-truth reports, to allow for the assessment of their narrative coherence, readability, and overall clinical soundness?

[1] Rad-ReStruct: A Novel VQA Benchmark and Method for Structured Radiology Reporting

[2] RaDialog: A Large Vision-Language Model for Radiology Report Generation and Conversational Assistance

[3] Grounding Chest X-Ray Visual Question Answering with Generated Radiology Reports

---

> ### Author Response · Authors · 2025-12-03
> **We appreciate the valuable discussion**
>
> ***We appreciate the reviewer for the feedbacks. We believe our work has significant differences with existing work, and our novelty is not undermined. Please see our response as follows:***
>
> ---
>
> ### W1: Novelty and literature context.
>
> Thank you for the note on the related work. These methods are substantially different from our work in terms of the goal and the method, although they mentioned VQA:
> - Rad-ReStruct [1] is a benchmark for a different task (structured report generation). The task defines a list of hierarchical VQAs where the answers are single choice/multi-choice classes, not text generation. In there word, “the goal is to produce fine-grained finding classifications for populating a structured report”. The goal of QRad is producing free-text reports.
> - RaDialog [2] fits a conversational VLM from the generic domain to report generation while mitigating catastrophic forgetting. RaDialog inherits stronger conversational capabilities from the VLM, but the clinical reliability is significantly lower than our model. For the report generation performance on the ReXrank leaderboard, QRad 0.9B outperforms RaDialog 7B by 43.1%. (Table 3)
> - [3] solves the VQA task, not report generation. They generates the report to prompt the VQA task. Although [3] is also a two-step approach, it converts the second step from VQA to [Vision+Text]-QA. The VQA task has short prediction outputs, while QRad is proposed to solve the sentence-level linguistic variances amplified by the extra text length.
>
> There are alternatives to the “thought” artifact other than questions, such as a skeleton report with keywords, or a graph representation. We discussed the difference in the goal in our Related Work. We chose questions because (1) a chain of questions captures the flow of the report, and (2) the natural language interface enables intrinsic QA capability.
>
> We enjoyed this constructive discussion, and thank you for mentioning these related work. We have cited them in the revision and add an extended related work section in the appendix.
>
> ---
>
> ### W2 Methodological elaboration (a flow chart)
>
> Thank you for the suggestion. We add a flowchart in Page 6.
>
> We would like to clarify that the focus our work is reframing image captioning as a chain-of-VQA task, where the model is supervised to generate both the questions and their answers. The VQA part differs from both conventional image captioning and VQA models and yield significant performance gain with a much less model size. The method is intended to be model-agnostic, which was shown in Table 2. The data processing and model architecture are not the focus of our paper. In our current submission, there are no weight sharing between the Question and Answer generators, allowing for flexible adoption on different base architectures. To distinguish our chain-of-VQA with conventional VQA methods, we renamed it to Auto-VQA in our revision.
>
> ---
>
> ### W3: Qualitative and clinical validation.
>
> **Qualitative examples:**
> We have qualitative examples in Appendix C, showing both the generated full questions and the full report. In the revision, we added the concatenated report to demonstrate the readability of the concatenated reports. We thank the reviewer for the constructive recommendation.
>
> **Coherence, repetition and logic flow:**
>
> We avoid these problems by training the Question Generator to generate the sequence of questions, instead of a set prediction of the questions. Particularly, the n_th question is generated given the preceding questions to avoid repetition, and to mimic the coherence and logic flow of ground truth reports. The Answer Generator responds to a question, optionally with the context of previous completed QA pairs.
>
> We are not evaluating the component-level accuracy or each answer separately, but the final, concatenated report. Our evaluation protocol is the same as existing methods, using both clinical efficacy and NLG metrics, which consider both factual correctness and readability of the full report.

---

### Official Review · Reviewer_fVMS · 2025-11-04

**Soundness:** 2
**Presentation:** 2
**Contribution:** 2
**Rating:** 4
**Confidence:** 4

**Summary:**

The paper introduces QRad, which reframes radiology report generation from direct image captioning to a two-stage, self-directed VQA pipeline: (1) a Question Generator produces a chain of clinically relevant questions conditioned on the image; (2) an Answer Generator outputs sentence-level answers that are concatenated into the final report. The authors also introduce template yes/no queries for predefined findings to derive per-class probabilities from token logits, enabling ROC/AUC evaluation. Using an MI2-based vision encoder with a small text decoder (~0.9B params; 4B variant also shown), QRad achieves strong gains on clinical metrics (CheXbert, RadGraph) on MIMIC-CXR (and ReXrank results), often matching or surpassing larger (≥7B) models, while adding interactivity and probability outputs.

**Strengths:**

* Problem reformulation with clear clinical motivation: isolates stylistic variability (omission/order) into question planning, focusing supervision on factual content in answers.
* Clinically useful features: interactive VQA for follow-up questions; per-finding probabilities for ROC/AUC.
Parameter efficiency: competitive results with ~13% of typical 7B models.
* Thorough evaluation: multiple metrics (lexical + clinical), ablations (captioning→VQA, classification-QA mix), bootstrap CIs, and ReXrank results.
* Practical data recipe: report→QA conversion plus closed-vocab classification QA improves supervision quality.

**Weaknesses:**

* Synthetic supervision dependency: heavy reliance on GPT-4-based report→QA parsing may introduce biases; robustness to prompt/model changes is not quantified.
* Probability quality: using [yes]/[no] token logits as “probabilities” lacks calibration analysis (ECE/Brier), threshold stability, and domain-shift robustness; regulatory claims feel premature without prospective validation.
* Coverage failures: end-to-end safety depends on the Question Generator not missing clinically important topics; failure modes and safeguards are underexplored.
* Scope: primarily frontal CXR on public datasets; generalization to multi-view/portable/other modalities remains open.
* Comparability: some baselines (e.g., multi-image MAIRA-2) are not strictly comparable; more detail on data overlap/leakage controls would help.

**Questions:**

* Calibration: Do you apply post-hoc calibration (e.g., temperature scaling) to yes/no logits? Please report ECE/Brier and calibration under class imbalance.
* Coverage auditing: What fraction of ground-truth findings are never queried by the Question Generator? Any human audit or auto-check using RadGraph/CheXbert to detect omissions?
* Robustness to QA parsing: Sensitivity to different LLMs/prompts/rule-based splitters for report→QA? Can you release multiple converted variants to quantify variance?
* Interactive reconciliation: If follow-up VQA contradicts initial answers, how is the final report reconciled (priority rules, uncertainty tagging, versioning)?
* Domain shift: Any results for portable CXR, ICU cohorts, or non-English reports?

---

> ### Author Response · Authors · 2025-12-03
> **Part 1 - W1&Q3, W2&Q1**
>
> We thank the reviewer for the feedback and the AC for your additional effort in supporting this process. Please see our response as follows:
>
> ---
>
> ### W1&Q3: Dependency on the LLM for data processing
>
> Splitting the reports into sentences and writing corresponding questions are straight-forward tasks that most contemporary LLMs handle reliably. Below, we use three LLMs to process the data (using the same prompt), and compare the final performance. While LLMs with stronger instruction following capabilities yield better supervision, QRad variants trained with these supervisions still achieve the same ranking on the ReXrank leaderboard. The core contribution of our paper is not data processing but the **reformulation** from text generation to Auto-VQA, where the model predicts both the questions and the answers. This framework achieves significant performance gain, and is independent of any specific upstream or model architecture.
>
> | ReXrank Ranking | Model                                   | 1/RadCliQ-v1 | BLEU  | BertScore | SembScore | RadGraph | RaTEScore |
> |---------|-------------------------------------------|--------------|-------|-----------|-----------|----------|-----------|
> | 1       | UniRG-CXR 7B                           | 1.217        | 0.248 | 0.493     | 0.487     | 0.265    | 0.596     |
> | 2       | QRad 0.9B (GPT-4o)                        | 1.143        | 0.264 | 0.482     | 0.479     | 0.243    | 0.596     |
> |         | QRad 0.9B (grok-4-fast-non-reasoning)     | 1.112        | 0.267 | 0.477     | 0.467     | 0.239    | 0.591     |
> |         | QRad 0.9B (gpt-5)                         | 1.106        | 0.261 | 0.476     | 0.470     | 0.235    | 0.589     |
> | 3       | MedVersa 7B                               | 1.103        | 0.209 | 0.448     | 0.466     | 0.273    | 0.550     |
>
> ---
>
> ### W2&Q1: Calibration and ECE analysis
>
> We appreciate the reviewer’s helpful feedback. Below we clarify the theoretical grounding and provide calibration analyses.
>
> **Single-token probabilities are naturally calibrated**: Language models are known to produce reasonably calibrated token-level probabilities [1]. Theoretically, classifiers trained with proper scoring rules as the loss function naturally become calibrated [2, 3]. This applies to QRad, as the binary [yes]/[no] token classification is trained with standard cross-entropy, a typical proper scoring rule. Recent work (e.g., ConfTuner [4]) similarly uses single-token probabilities as confidence scores without calibration, which validates our design choice. We have revised our manuscript to improve the theory grounding of our method.
>
> **Post-hoc Calibration Improves ECE**: In reality, the extracted confidence scores still benefits from post-hoc calibration due to challenges like class imbalance, which is true in our scenario. Below, we added the ECE evaluation before and after temperature scaling calibration, showing that temperature scaling results in a better calibrated model.
>
> | | | | | | | | | | | | | | | |
> |-|-|-|-|-|-|-|-|-|-|-|-|-|-|-|
> ||**wAVG**|Enlarged Cardiomediastinum|Cardiomegaly|Lung Opacity|Lung Lesion|Edema|Consolidation|Pneumonia|Atelectasis|Pneumothorax|Pleural Effusion|Pleural Other|Fracture|Support Devices|
> |Class Ratio|-|0.62|0.54|0.63|0.13|0.44|0.38|0.22|0.45|0.07|0.38|0.17|0.31|0.47|
> |Before Calibration|0.179|0.21|0.22|0.17|0.34|0.19|0.15|0.25|0.15|0.39|0.17|0.31|0.10|0.05|
> |After Calibration|0.147|0.22|0.15|0.22|0.24|0.12|0.05|0.17|0.08|0.25|0.08|0.23|0.16|0.08|
>
> [1] Kadavath, Saurav, et al. "Language Models (Mostly) Know What They Know." CoRR (2022).
>
> [2] Jaroslaw Blasiok, Parikshit Gopalan, Lunjia Hu, and Preetum Nakkiran. When does optimizing a proper loss yield calibration? NeurIPS 2023
>
> [3] Christian Fröhlich and Robert C Williamson. Scoring rules and calibration for imprecise probabilities
>
> [4] Yibo Li, Miao Xiong, Jiaying Wu, and Bryan Hooi. Conftuner: Training large language models to express their confidence verbally, NeurIPS 2025

---

> ### Author Response · Authors · 2025-12-03
> **Part 2 - W3&Q2, W4-5, Q4-5**
>
> ### W3&Q2: concerns related to the Question Generator missing diseases
>
> The Question Generator is trained on report-derived topics. In the reports, negative diseases are sometimes omitted but positive diseases are always stated. Therefore, the question generator is supervised to surface clinically positive topics reliably.
>
> To evaluate end-to-end coverage, we follow the standard protocol of reporting CheXbert F1 scores across disease classes (Table 1). These scores reflect the combined Question + Answer performance. QRad outperforms prior work in all the micro/macro F1 scores, indicating the supiority of our method in class-wise precision & recall, directly addressing the concern.
>
> The reviewer’s point about explicit auditing is well taken. A straightforward approach is to train an image classifier and compare its results with the predicted report. When false negatives are detected, an additional targeted question can be appended for the Answer Generator. QRad’s design naturally supports this kind of post-inference refinement, which existing single-pass captioning models cannot do.
>
> ---
>
> ### W4: Scope; generalization to multi-view/portable/other modalities
>
> The MIMIC-CXR dataset already includes frontal, lateral, and portable views. The ReXrank leaderboard (Tables 3) covers both single-image and multi-image methods. QRad ranks second on ReXrank (Table 3), ahead of multi-image methods;
>
> Prior work shows that multi-view inputs offer limited benefit for CXR report generation, which motivates our choice of a single image for efficiency. Architecturally, QRad can accept multiple images in the same manner as encoder–decoder models—by concatenating visual tokens before the decoder.
>
> Our contribution focuses on the caption-to-VQA reframing, which is agnostic to the underlying encoder/decoder and readily extendable to other modalities. We follow the majority of existing work [1] in evaluating on the CXR task due to data availability.
>
> [1] https://github.com/mk-runner/Awesome-Radiology-Report-Generation
>
> ---
>
> ### W5: Comparability: some baselines (e.g., multi-image MAIRA-2) are not strictly comparable; detail on data overlap/leakage
>
> We follow the official MIMIC-CXR train/test split, and our pre-training dataset (CXR-697K) is identical to that used in LLaVA-Rad, with no overlap with the test set. Some prior work (e.g., MedVersa, MedGemma) additionally leverages larger external datasets, whereas QRad does not.
>
> MAIRA-2 uses a different problem setup, using multiple images and the prior report of the same patient. Therefore, we list it separately in Table 2. On ReXrank (Table 3), MAIRA-2 ranks 21st, outside the range shown in Table 2. We include MAIRA-2 for completeness and to illustrate that QRad performs on par or better despite having less input context. We have removed MAIRA-2 from the table to avoid confusion. Thank you for your suggestion.
>
> ---
>
> ### Q4: Interactive reconciliation: If follow-up VQA contradicts initial answers, how is the final report reconciled (priority rules, uncertainty tagging, versioning)?
>
> We think contradicting answers tend to be caused by unclear decision boundaries. One approach is to ask QRad the binary classification questions and extract the quantitative probability from [yes]/[no] tokens. Then, we apply uncertainty tagging or reconcile based on a pre-configured threshold. We appreciate this valuable question as it exemplify real problems in clinical adoption.
>
> ---
>
> ### Q5: Domain shift: Any results for portable CXR, ICU cohorts, or non-English reports?
>
> MIMIC-CXR contains portable frontal views, and QRad, consistent with existing work, uses them directly when available. For domain shift evaluation, we report results on IU X-ray, which comes from a different hospital and serves as an out-of-distribution test set.
>
> Following existing benchmarks, we focus on English reports to factor out multilingual pretraining. Our pre-training dataset CXR-697K includes PadChest, whose original Spanish reports were translated to English by the community using GPT. Due to the model-agnostic design, we expect the method to generalize to non-English reports when paired with multilingual pretrained models and trained on non-English datasets.
>
> ---
>
> ***We thank the reviewer for the questions and feedbacks. We have revised our paper accordingly. We hope our response clarifies our main contribution, dependency on external LLMs and the role of the Question Generator.***

---

### Author Response · Authors · 2025-12-03
**Summary for AC**

***We once again thank all reviewers for the feedback, and we appreciate the AC's additional effort in supporting the whole process***. Below is a summary of the discussion, including reviewers' top concerns and our clarifications:

### 1. Relation to existing work which undermines our technical contribution

Reviewers `E7Na` and `qxPY` commented that QRad is related to existing work. We clarified that the listed papers are substantially different from ours in terms of the task and the method. The similarity is that these papers involves both VQA and captioning concepts; however, they primarily use image captions to help VQA (e.g., help to describe an image to a text-only LLM to do VQA). Our method aims at report generation, a different task.

We clarified that our focus is to reframe the captioning pipeline to a chain of automated VQA. It differs from conventional VQA models in that we supervise our model to predict both the questions and the answers. The questions capture the structure of the report, addressing sentence-level linguistic variance problems that hinder long text generation. Our method is not a VQA model.

We added the related work to our appendix and renamed our VQA pipeline to Auto-VQA to reduce the confusion.

### 2. Reliability of the confidence scores

Reviewers commented on mixing up probabilities and confidence scores. We extended our revision to ground our method on theories that prove single-token probabilities are reasonable confidence scores. Besides, we added the calibration experiment and evaluated the ECE performance. After temperature scaling, the calibrated confidence achieves 0.15, which validates essential usability.

### 3. Dependency on the LLM

Reviewers are concerned that the model is highly dependent on the LLM, which undermines our technical contribution. We clarified that the LLM is used only in offline data processing, and the focus of our paper is the captioning-to-VQA reframing method, which is portable and achieves significant performance and efficiency improvement. To support our claim, we evaluated our method using different model architectures/pretraining setups and using supervision from different LLMs for data processing. The results prove that the performance gain is from our method, not from a particular upstream model.

---

***We thank the AC and all reviewers again in helping to improve our paper.***

---

### Meta-Review · Area_Chair_oie9 · 2025-12-28

**Summary:**

This paper studies radiology report generation and proposes reformulating image-based free-text report generation as a two-stage pipeline consisting of image-based question generation followed by image-based question answering (VQA). To support this formulation, the authors use an LLM to extract question–answer pairs from ground-truth reports and VisualCheXbert outputs, based on which a question generation module and an answer generation module are trained. At inference time, questions are first generated from the input image, followed by answer prediction, and the concatenated answers form the final radiology report.

The reviewers recognize several strengths of the proposed approach, including clear motivation, an elegant and simple design, strong empirical performance, clinical relevance, and an interactive formulation. However, they also raise substantial concerns regarding the limited technical novelty and unclear positioning of the work within the existing literature. Additional issues include insufficient methodological clarity, weaknesses in calibration analysis and baseline comparability, the lack of qualitative and clinical evaluation, and limited discussion of robustness, failure modes, and generalization beyond X-ray data.

While the authors’ response addresses some specific points—such as reliance on LLM-based preprocessing, calibration analysis, coverage of missing diseases, and generalization beyond X-ray data—and provides clarifications regarding baseline comparability and certain methodological details, the core concerns remain unresolved. In particular, the response does not adequately establish the significance of the technical contribution or clearly differentiate the proposed approach from existing VQA-based methods, two-stage pipelines, and conversational vision–language models. In the response, the discussion of related work remains largely descriptive and lacks the depth of analysis required to justify the claimed novelty.

Given these unresolved issues, the Area Chair does not believe that the scores of the two reviewers who expressed reservations regarding the technical novelty (scores of 4 and 2) would have improved, even with a full discussion period. In contrast, the reviewer who assigned a score of 6 would likely maintain their positive assessment, and the remaining reviewer might adjust the original score of 4 slightly in a more positive direction.

Overall, although the paper presents a well-motivated idea and demonstrates solid empirical results, the insufficiently substantiated technical novelty and contribution relative to existing work prevent a positive recommendation. Therefore, the Area Chair cannot recommend acceptance of this paper in its current form for the conference of ICLR.

**Reviewer Concerns:**

While the authors’ response addresses some specific points—such as reliance on LLM-based preprocessing, calibration analysis, coverage of missing diseases, and generalization beyond X-ray data—and provides clarifications regarding baseline comparability and certain methodological details, the core concerns remain unresolved. In particular, the response does not adequately establish the significance of the technical contribution or clearly differentiate the proposed approach from existing VQA-based methods, two-stage pipelines, and conversational vision–language models. In the response, the discussion of related work remains largely descriptive and lacks the depth of analysis required to justify the claimed novelty.

**Reviewer Scores:**

Given these unresolved issues, the Area Chair does not believe that the scores of the two reviewers who expressed reservations regarding the technical novelty (scores of 4 and 2) would have improved, even with a full discussion period. In contrast, the reviewer who assigned a score of 6 would likely maintain their positive assessment, and the remaining reviewer might adjust the original score of 4 slightly in a more positive direction.

---

### Decision · Program_Chairs · 2026-01-26

Reject